# An Overview of the Epigenetic Modifications in the Brain under Normal and Pathological Conditions

**DOI:** 10.3390/ijms25073881

**Published:** 2024-03-30

**Authors:** Laura Lossi, Claudia Castagna, Adalberto Merighi

**Affiliations:** Department of Veterinary Sciences, University of Turin, Largo Paolo Braccini 2, 10095 Grugliasco, Italy; laura.lossi@unito.it (L.L.); claudia.castagna@unito.it (C.C.)

**Keywords:** epigenetics, DNA, histones, brain, neurons, development, neurodegeneration

## Abstract

Epigenetic changes are changes in gene expression that do not involve alterations to the DNA sequence. These changes lead to establishing a so-called epigenetic code that dictates which and when genes are activated, thus orchestrating gene regulation and playing a central role in development, health, and disease. The brain, being mostly formed by cells that do not undergo a renewal process throughout life, is highly prone to the risk of alterations leading to neuronal death and neurodegenerative disorders, mainly at a late age. Here, we review the main epigenetic modifications that have been described in the brain, with particular attention on those related to the onset of developmental anomalies or neurodegenerative conditions and/or occurring in old age. DNA methylation and several types of histone modifications (acetylation, methylation, phosphorylation, ubiquitination, sumoylation, lactylation, and crotonylation) are major players in these processes. They are directly or indirectly involved in the onset of neurodegeneration in Alzheimer’s or Parkinson’s disease. Therefore, this review briefly describes the roles of these epigenetic changes in the mechanisms of brain development, maturation, and aging and some of the most important factors dynamically regulating or contributing to these changes, such as oxidative stress, inflammation, and mitochondrial dysfunction.

## 1. Introduction

Understanding the cellular and molecular mechanisms underlying aging is of paramount importance due to the profound impact that aging has on human health and society. From this perspective, epigenetics provides a crucial framework for comprehending its molecular underpinnings. The genetic code, inscribed within the DNA sequence, serves as the blueprint for life. Yet, within each cell, an additional layer of information, the epigenetic code, dictates which and when genes are activated, thus orchestrating gene regulation and playing a central role in development, health, and disease.

Epigenetics, a term coined by the British developmental biologist Conrad Waddington in the mid-20th century [1], is the study of changes in gene expression that do not involve alterations to the underlying DNA sequence. Epigenetic changes can be influenced by a variety of factors, which may be intrinsic to the organisms, such as developmental processes, but also extrinsic, i.e., environmental exposures and lifestyle choices. Remarkably, various extrinsic factors have been recognized as potential modifiers of epigenetic patterns, including nutrition, obesity, physical activity, tobacco smoking, alcohol use, exposure to environmental contaminants, psychological stress, and working night shifts [2,3]. Over time, epigenetic modifications can accumulate, leading to the establishment of an “epigenetic landscape” unique to an individual’s aging process. During aging, changes in these epigenetic marks can lead to alterations in gene expression patterns, contributing to age-related phenotypes and diseases. Among the several types of epigenetic modifications, specific DNA methylation patterns correlate with chronological age. On these premises, researchers have developed a series of so-called epigenetic clocks that provide a molecular measure of aging and can be used to assess the biological age of an individual, which may differ from their chronological age [4]. Understanding the epigenetic changes associated with aging opens avenues for potential interventions to slow down or reverse age-related conditions [5,6]. Starting from this understanding, epigenetic therapies are being explored to rejuvenate tissues and combat age-related diseases [7,8].

### 1.1. Literature Search Strategy

If one searches for the string “epigenetic changes in the brain” in PubMed, filtering for reviews or systematic reviews in the last ten years, more than 1200 papers are retrieved. In writing this paper, we aimed to focus on the studies describing, histologically and functionally, the main epigenetic modifications in the mammalian brain during development, adulthood, and old age. We also took into consideration the more important findings related to the most diffused neurodegenerations, i.e., Alzheimer’s disease (AD) and Parkinson’s disease (PD). We used PubMed as a starting database but also made direct Internet searches of exact phrases (in double quotes) related to the most important topics to be addressed. The keywords used were “epigenetic”, “brain”, “neurons”, “development”, “aging”, and “mammals”. These terms were first identified in the PubMed database and their synonyms were recognized in the thesaurus. Variations in search terms were also considered. The search was primarily focused on the last 10 years, but older relevant papers were also included.

### 1.2. Types of Epigenetic Modifications

Today, there is no consensus in the field regarding the definition of epigenetic modifications. In his Nature paper of 2007 entitled Perceptions of Epigenetics, Adrian Bird emphasized that, for epigeneticists, there is no obvious ‘epigene’ [9]. He proposed defining epigenetics as the structural adaptation of chromosomal regions to register, signal, or perpetuate altered activity states, by this means implicitly depicting epigenetic markings as reactive rather than proactive. Later, in 2008, a different definition was proposed in a meeting on chromatin-based epigenetics hosted by the Banbury Conference Center and Cold Spring Harbor Laboratory, defining an epigenetic trait as a stably heritable phenotype resulting from changes in a chromosome without alterations in the DNA sequence [10]. Thus, there is no consensus as to whether or not non-coding RNAs could be regarded as participating in epigenetics. Yet Shelley Berger and colleagues, in providing their view and interpretation of the proceedings at the meeting, considered non-coding RNAs among epigenetic initiators [10]. Since non-coding RNAs, as discussed later in this paper, have been described as participating in several brain epigenetic modifications, we will consider them as a third category of these modifications.

There are three main groups of epigenetic modifications (Figure 1). DNA methylation and histone modifications are described initially. These two processes affect gene expression by acting on the chemical structure of the DNA or the histones, respectively. In more recent times, many modification processes of RNAs leading to the synthesis of so-called non-coding RNAs were acknowledged to introduce a new level to the gene regulation process, resulting in novel RNA epigenetics [11].


*DNA Methylation:*


The addition of a methyl group to a cytosine base in the DNA, typically occurring in the context of CpG dinucleotides, is known as DNA methylation [12,13]. CpG DNA methylation plays a crucial role in regulating gene expression and often leads to gene silencing. DNA methylation is a primordial mechanism observed in all realms of life. While the enzymes responsible for methylation have stayed mostly the same, DNA methylation has evolved to serve many purposes over time, such as protecting against transposable elements and regulating gene expression. Human disorders, including neurodegenerative diseases, are associated with abnormalities in DNA methylation [14]. CpG sites, also known as CG sites, refer to certain regions inside the DNA molecule where a cytosine nucleotide is immediately followed by a guanine nucleotide in the linear arrangement of bases along its 5′ → 3′ orientation. CpG sites are found at a high frequency inside genomic areas known as CpG islands, also referred to as CG islands. CpG islands are characterized by having a GC content of over 50%, an observed CpG ratio (Obs/Exp) greater than 0.6, and a length of over 200 base pairs. When CpG sites cluster into islands, they are generally protected from DNA methylation. However, some islands within genes (referred to as intragenic CpG islands) defy this repression and remain transcriptionally active. Intragenic CpG islands influence mRNA isoform length, thereby expanding transcriptome diversity [15]. Remarkably, the differential DNA methylation of CpG islands in normal human adult somatic tissues allows for the distinction between neural and non-neural tissues [16], and a database of DNA methylation profiles in the brain has been published [17]. DNA methylation acts as a molecular “off switch” by preventing transcription factors and RNA polymerase from accessing the promoter region of the gene, thereby inhibiting gene expression [18]. DNA methyltransferases (DNMTs), methyl-CpG binding proteins (MBPs), and ten-eleven translocation proteins enable the maintenance, interpretation, and removal of DNA methylation [19] (Table 1). Different forms of methylation, including 5-methylcytosine (5 mC), 5-hydroxymethylcytosine, and other oxidized forms, have been detected by recently developed sequencing technologies. Another form of DNA methylation occurs on adenine. In DNA biology, only three forms of adenine methylation are considered physiologically significant. These compounds, 1-methyladenine (1 mA) and 3-methyladenine (3 mA), are considered by many to be the result of alkylation damage in DNA and, thus, are significant in the study of genome stability and DNA repair [20]. Recent developments in detection methods have shown DNA N6-methyl deoxyadenosine (6 mA) as a methylation alteration at the sixth position of adenine in DNA, with an impact on neurodegeneration [21].


*Histone Modifications:*


Histone modifications are central regulators of gene expression, as they determine which genes are turned on or off in each cell or tissue, affecting cellular function [22] and, in the general frame of this paper, contributing to the aging phenotype. Histones are protein spools around which DNA is wound. Chemical modifications, such as acetylation [23], methylation [24], lysine di-methylation and tri-methylation [25,26], phosphorylation [27], ubiquitination [28], sumoylation [29], lactylation [30], serotonylation [31], and crotonylation [32] can alter the structure of histones and, consequently, regulate DNA transcription. The interplay of these modifications at specific histone residues creates a dynamic “chromatin landscape” that can either promote or inhibit gene transcription. Thus, by modifying the chromatin structure and accessibility, the different types of histone modifications provide a complex regulatory framework that governs gene expression. The precise combinations of these modifications at specific histone residues create a “histone code” that can be read and interpreted by various cellular machinery to dictate gene transcription outcomes [33].


*Non-coding RNAs (ncRNAs):*


ncRNAs are functional RNA molecules transcribed from DNA but not translated into proteins. They were initially thought to regulate gene expression, primarily at the post-transcriptional level. However, recent research has revealed that various classes of ncRNAs participate in epigenetic control. Small RNA molecules, like microRNAs (miRNAs) and long non-coding RNAs (lncRNAs), can bind to messenger RNAs (mRNAs) and inhibit their translation or promote their degradation. This post-transcriptional gene regulation is a fundamental part of epigenetic control [34].

#### 1.2.1. DNA Methylation

Remarkably, the brain exhibits a notable concentration of DNA methylation, but it is important to note that 5 mC only constitutes around 1% of the total nucleic acids present in the mammalian genome, which, in general, shows a scarcity of CpG sites. CpG sites that are observed across the genome exhibit a significant level of methylation, except in CpG islands. It is noteworthy that non-CpG methylation has been observed in both mouse and human embryonic stem cells. However, it is important to note that this methylation is not present in mature tissues [12].

**Table 1 ijms-25-03881-t001:** Proteins acting on DNA methylation in the nervous tissue. For abbreviations see the list at the end of the main text.

Protein Family	Family Members	Main Functions
**DNMTs**	DNMT1 DNMT3A DNMT3B	DNMT1 is the switch from neurogenesis to gliogenesis during NSC differentiation [35]. DNMT3A regulates NSC proliferation and differentiation [36] and controls adult hippocampal neurogenesis of GABAergic neurons [37]. DNMT3B is required for the proper timing of neuronal differentiation and maturation [38].
	*MBD proteins* MBD1–5 2MeCP2	MBD1 deficiency causes the accumulation of NSCs and the impairment of neuronal lineage differentiation [39,40,41] and contributes to the genesis of acute pain by epigenetic gene silencing in primary sensory neurons [42]. MBD2 and MBD3 are crucial for ESC differentiation to neural cells [43]. MBD4 intervenes in selective gene expression profiles in cortical neurons [44]. MBD5 controls neurite outgrowth and is responsible for 2Iq23.1 microdeletion syndrome [45]. 2MeCP2 controls neuronal maturation and dendritic arborization during development [46] and in the adult brain [47].
**MBPs**	*Zinc finger*/*Kaiso proteins* Kaiso/ZBTB33 ZBTB4 ZBTB38	Kaiso/ZBTB33 intervenes in the neuronal commitment of NSCs [48]. ZBTB4 controls gene expression in different types of neurons (hippocampus, olfactory pathways, motor nuclei of the brainstem, and granular layer of the cerebellum) [49] and is associated with age-at-onset AD [50]. ZBTB38 can repress transcription by binding to methylated DNA. It leads to early embryonic death via the suppression of the transcription factors Nanog and Sox2 [51].
	*SRA proteins* UHRF1 UHRF2	UHRF1 is critical for the maintenance of DNA methylation through cell division and is involved in DNA damage repair. It regulates the proliferation of NSCs [52]. UHRF2 is involved in cell cycle progression and controls the transition from RPCs to differentiated cells by regulating the cell cycle, epigenetic modifications, and gene expression [53].

Recent research has provided a more comprehensive examination of the murine frontal cortex. This investigation has demonstrated that, while the bulk of methylation events take place at CpG sites, a notable proportion of methylation also happens at non-CpG sites.

Significantly, the impact of DNA methylation on gene activity can vary depending on the specific genomic areas and the underlying genetic sequence. DNA methylation can occur in intergenic regions, CpG islands, and the gene body [12,54,55,56,57]. An estimated 45% of the mammalian genome comprises transposable and viral elements, which are rendered inactive by the process of bulk methylation in the DNA intergenic regions. The potential danger of these elements lies in their ability to cause gene disruption and DNA mutation when replicated and inserted. A significant proportion of gene promoters (frequently of housekeeping genes), around 70%, are located within CpG islands. CpG islands, particularly those linked to promoters, exhibit a significant degree of conservation across both mice and humans. The presence and conservation of CpG islands over evolutionary time suggest that these genomic areas hold significant functional significance. It has been possible to predict the methylation status of CpG islands in the human brain using a bioinformatic approach reaching an 84.65% specificity and an 84.32% sensitivity [58]. An examination of DNA methylation conservation between humans and mice revealed that there is no correlation between DNA methylation and sequence conservation. On the contrary, there is a positive correlation between a higher CpG density and a greater conservation of DNA methylation. Furthermore, the conservation and alteration markers of DNA methylation throughout mammalian brain evolution are strongly influenced by genomic context, in addition to CpG abundance [59]. Remarkably, when the CpG methylation landscapes of adult mouse neurons in the hippocampal dentate gyrus before and after synchronous neuronal activity were compared, approximately 1.4% of the 219,991 CpGs that were assessed exhibited fast active demethylation or new methylation [60]. Certain alterations stayed unchanged for a minimum of 24 h. The CpGs that were modified by the activity showed a wide distribution across the genome, with notable concentrations in places with a low CpG density. These modified CpGs were linked to genes that are particular to the brain and are involved in neural plasticity [60]. The gene body is commonly defined as the region of the gene that extends beyond the initial exon, as it has been observed that methylation of the first exon, like promoter methylation, can result in the suppression of gene expression. Multiple studies have provided evidence indicating that the process of DNA methylation occurring within the gene body is positively correlated with an increased level of gene expression in cells undergoing division [61]. Nevertheless, in cells that divide at a slow rate or do not divide at all, such as those found in the brain, it has been observed that gene body methylation does not correlate with the upregulation of gene expression [12]. Many genes exhibiting unique methylation patterns specific to certain cell types were detected after a DNA methylation analysis on purified neurons and glia from postmortem human brain tissues [62]. Specifically, distinct changes in methylation patterns related to aging, particularly in neurons, such as *CLU*, *SYNJ2*, and *NCOR2*, as well as in glia, including *RAI1*, *CXXC5*, and *INPP5A*, were observed. In addition, unique connections between neurons or glial cells and the progression of AD Braak stages were discovered in genes such as *MCF2L*, *ANK1*, *MAP2*, *LRRC8B*, *STK32C*, and *S100B*. DNA methylation has also been involved in the epigenetics of aging [63]. Studies on anti-aging therapies in mice, such as caloric restriction, dwarfism, and rapamycin treatment, provide the most compelling evidence that age-related alterations in DNA methylation contribute to the aging process. These anti-aging treatments slow down epigenetic clocks and can correct or prevent from 20 to 40% of age-related alterations in DNA methylation [54].

#### 1.2.2. Histone Epigenetic Modifications

Histones are proteins found in the cell nucleus that play a critical role in packaging and organizing DNA into a compact structure called chromatin. Epigenetic modifications of histones involve chemical alterations to these proteins, which can have profound effects on gene expression and, consequently, various cellular processes. These modifications are essential for the regulation of gene activity and have far-reaching implications in development, health, and disease.

Several key histone modifications have been extensively studied. These include acetylation, methylation, phosphorylation, ubiquitination, sumoylation, lactylation, crotonylation, and serotonylation (Table 2 and Figure 1 and Figure 2).

*Acetylation*, the addition of an acetyl group, typically on lysine residues on histone tails, is generally associated with gene activation. Acetyl groups are added to histones by histone acetyltransferases (HATs). This process neutralizes the positive charge on histones, leading to a relaxed chromatin structure that is more open and accessible to transcription factors and RNA polymerase [64]. The ensuing changes affect the accessibility of DNA to cellular machinery [65].

*Methylation*, the addition of 1–3 methyl groups to lysine or arginine residues on histone proteins, is carried out by histone methyltransferases (HMTs) [66] and can be associated with both gene activation (by recruiting chromatin-modifying complexes) and repression (forming barriers to inhibit transcription), depending on the specific histone residue and the degree of methylation [24].

*Phosphorylation* is associated with changes in chromatin structure during various cellular processes, including DNA replication, repair, and mitosis. Histone phosphorylation is the addition of phosphate groups to serine, tyrosine, or threonine residues on histone tails. This modification can alter the chromatin structure and facilitate gene activation or repression [67].

*Ubiquitination*, the addition of ubiquitin molecules to specific lysine residues on histone tails, is carried out by ubiquitin ligases. Ubiquitination can affect gene expression, as it alters the chromatin structure by recruiting proteins that either activate or repress transcription. It plays a role in transcriptional regulation and DNA repair [68].

*Sumoylation*, the addition of small ubiquitin-like modifier (SUMO) proteins to specific lysine residues on histone tails, is catalyzed by SUMO ligases. Sumoylation can affect the chromatin structure and gene expression by recruiting proteins that modulate transcriptional activity and contribute to genome stability [69].

*Lactylation* is the addition of lactate to histone molecules. Lactylation can accelerate transcription and promote gene expression. It has been implicated in several disease model molecules [70].

**Table 2 ijms-25-03881-t002:** Histone modifications and their biological effects. Amino acid residues are indicated by one-letter notation [71]. Abbreviations: Ac = acetylation; Cr = crotonylation; La = lactylation; Me = methylation; P = phosphorylation; Ser = serotonylation; Su = sumoylation; and Ub = ubiquitination.

Histone	Type of Modification	Residue(s)	Biological Effect
**H1**	Su	K17, K21, K34	Gene repression, chromatin compaction, and restriction of embryonic cell fate identity [72]
**H2A**	Ac	K5	Gene activation [73]
	P	S1	Mitosis [74]
	P	T120	Mitosis, gene activation [27]
	Su	K127	Gene repression, chromatin compaction [75]
	Ub	K119	Gene repression [76]
**H2AX**	P	S139	DNA repair [77]
	Su	K5, K9, K13, K15, K118, K119, K127, K133, K134	Gene repression, chromatin compaction [78]
**H2B**	Ac	K5, K12, K15, K20	Gene activation [79]
	P	S14	Apoptosis [80,81]
	Su	K16	Gene repression, chromatin compaction [82]
	Ub	K12	Gene activation [83]
**H3**	Ac	K4, K9, K14, K18, K23, K27, K36	Gene activation [84]
	Ac	K56	Histone deposition [85]
	Cr	K9	DNA repair [32]
	Cr	K4, K14, K18, K27	Gene activation [86]
	La	K4, K18, K79	Gene activation [30]
	Me	K9, K27	Gene repression [87]
	Me	R2, R8, R17, R26	Gene activation [88]
	P	T6	Gene activation [89]
	P	S10, S28, T3, T11	Mitosis, DNA repair [90,91]
	P	T45	DNA replication, response to DNA damage [92]
	Ser	Q5	Gene activation [93]
	Su	K18	Gene repression, chromatin compaction [94]
	Ub	K23	Maintenance of DNA methylation [95]
**H4**	Me	R3	Gene activation [96]
	P	S1	Mitosis, gene activation [74]
	Ac	K12, K91	Histone deposition [97]
	Ac	K5, K8, K12, K16	Gene activation [98]
	Me	K20	Gene repression [99]
	Su	K5, K8, K12, K16, K20	Gene repression, chromatin compaction [100]

*Crotonylation* is the addition of crotonyl groups to histone lysine residues. Crotonylation plays a role in DNA damage and repair, and gene activation [101].

*Serotonylaton* was discovered in 2019 when it was demonstrated that serotonin can be covalently attached to histone H3 by transglutaminase 2 (TGM2) [102]. The identification of histone serotonylation uncovered a fascinating prospect in which small molecules engaged in intercellular communication might be directly associated with chromatin, introducing an extra layer of intricacy to both chromatin regulation and neurotransmitter-dependent signaling networks [103].

**Figure 2 ijms-25-03881-f002:**
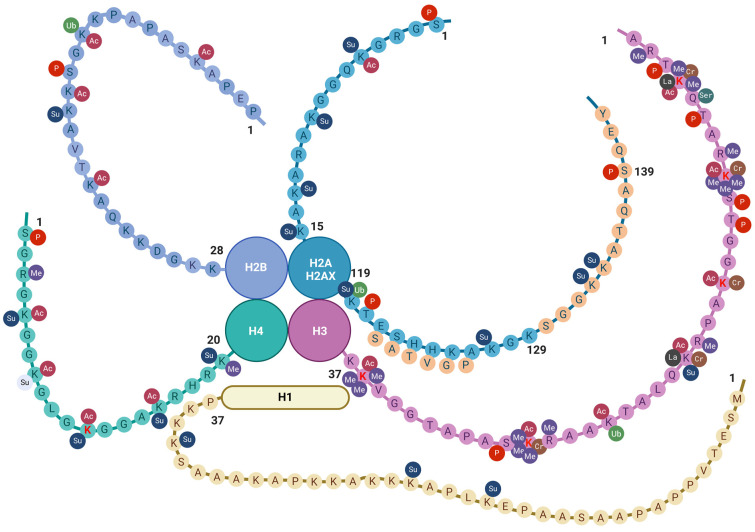
Histone tail epigenetic modifications described in the brain. All core histone proteins contain intrinsically disordered tail regions that protrude from the DNA-enveloped core and are known to play critical roles in chromatin regulation (see text). The linker histone H1 has a three-domain structure consisting of a short N-terminal tail, a central globular domain, and a long, extremely basic C-terminal tail [104]. Only the N-terminal tail is shown here. Aminoacidic residues are indicated by one-letter notation [71]. Amino lysine acid residues (K) undergoing epigenetic changes so far described in the old brain are indicated in red font. Sequences specific to H2AX are indicated by light orange circles. Abbreviations: AC = acetylation; Cr = crotonylation; Las = lactylation; Me = methylation; P = phosphorylation; Ser = serotonylation; Su = sumoylation; and Ub = ubiquitination. Created with BioRender.com.

The above histone modifications are pivotal for gene regulation and genome stability. They contribute to various cellular processes, including gene expression, cell differentiation, and DNA repair, and, thus, are crucial in cell fate determination during development, contributing to the establishment of lineage-specific gene expression patterns. Remarkably, some histone modifications can be passed on to daughter cells during cell division, contributing to the so-called “epigenetic inheritance” [105]. Epigenetic inheritance was first described in mice upon the demonstration of a modification of the Kit gene in the progeny of heterozygotes with the null mutant Kit(tm1Alf) and relevant loss-of-function pigmentation phenotypes, affecting adult phenotypes in multiple following generations of mice. This gene modification was associated with the zygotic transfer of RNA molecules, similar to the paramutation induced in plants by cross-talk between allelic loci [106]. Later, the same group demonstrated that the paramutation mechanism was of relevance to pathophysiology by injecting fertilized mouse eggs with RNAs targeting Cdk9, a key regulator of cardiac growth, and obtaining cardiac hypertrophy in the progeny [107].

#### 1.2.3. ncRNAs

ncRNAs that do not undergo translation to produce proteins can be categorized into two main groups: housekeeping ncRNAs and regulatory ncRNAs [108,109,110]. RNA molecules with regulatory functions can be broadly classified into two main categories according to their size: short-chain non-coding RNAs, which encompass small interfering RNAs (siRNAs), miRNAs, and PIWI-interacting RNAs (piRNAs), and lncRNAs [109,111]. siRNAs have a size of 19–24 bp, derive from double-stranded DNA, and silence gene transcription [112]. The same function is accomplished by miRNAs that are 19–24 bp long and originate from hairpin-containing primary transcripts (pri-miRNA) [112]. Notably, miRNAs have been associated with the regulation of neural stem cell (NSC) differentiation, apoptosis, and some neurodegenerative disorders [113,114]. piRNAs are of a larger size (26–31 bp), derive from long-chain size precursors, and repress transposons via transcriptional or post-transcriptional mechanisms [115]. lncRNAs have a size of more than 200 bp, derive from multiple sources, and regulate gene expression in various ways, including epigenetic, transcriptional, post-transcriptional, translational, and protein location mechanisms [109]. Remarkably lncRNAs have been implicated in the early response of neurons to BDNF stimulation [116].

## 2. Epigenetic Regulation in the Developing and Mature Brain

Epigenetic modifications are central players in the intricate processes of brain development and function. The dynamic interplay between genetics and epigenetics shapes the complexity of the brain and its ability to adapt to a constantly changing environment. Broadly speaking, epigenetic and epitranscriptomic changes, i.e., the RNA editing that affects mRNA functions, regulates neuronal lineage, differentiation, and connectivity, with obvious consequences on the structure and function of synapses. In addition, both types of changes have been associated with several neurodevelopmental disorders [117]. Interestingly, the dysregulation of epigenetic processes has been implicated in autism spectrum disorder (ASD) and intellectual disabilities, further highlighting the significance of epigenetics in brain development and maturation [118].

In the developing mammalian cortex, radial glial cells (RGCs) act as primary NSCs and give rise to a variety of neurons and glial cells following intricate developmental programs with astounding spatiotemporal accuracy. Controlling RGCs’ temporal competency is a crucial mechanism for the cerebral cortex’s highly preserved and predictable structure. Remarkably, the pattern of gene expression of RGCs is largely shaped by several epigenetic controls, including DNA methylation, histone modifications such as H3K4me3 and H3K27me3, and 3D chromatin architecture [119]. Epitranscriptomic changes, such as m^6^A-eRNA methylation and m5C RNA methylation, also control the function and turnover of cell-type-specific transcripts, which, in turn, regulates the temporal pre-patterning of RGCs [119]. DNA methylation patterns have a well-recognized role in NSC proliferation and differentiation and help to establish and maintain the specific identity of neurons, contributing to the diversity of neuronal subtypes in the brain [19,120]. It has been demonstrated that, during the transition from fetal to young adult development, there is a significant rearrangement of the methylome, which is closely associated with the process of synaptogenesis. During this temporal phase, there is a notable accumulation of highly conserved non-CG methylation (mCH) specifically in neurons, but glial cells do not exhibit a similar pattern [120]. Consequently, mCH emerged as the prevailing form of methylation within the human neuronal genome. Other studies have provided comprehensive and high-resolution maps of 5-hydroxymethylcytosine (hmC) at the single-base level. These maps have revealed that hmC is present in the genomes of fetal brain cells, specifically marking sites that are believed to be involved in regulatory processes [120]. Growing evidence also suggests that DNA cytosine and hydroxyl cytosine methylation carried by DNMTs and/or MBPs plays a pivotal role in neurogenesis, neuronal differentiation, synaptogenesis, learning, and memory [121] (Table 2). It has also been recently demonstrated that DNA methylation regulates the differentiation of oligodendrocytes and Schwann cells during development and repair [122]. Remarkably, experience-dependent DNA methylation can modify gene expression and contribute to the brain’s ability to adapt to environmental challenges [123]. On the other hand, the mechanisms of deviant DNA methylation in neurodegenerative diseases continue to be unclear. Remarkably, DNA methylation modified and potentially restored youthful gene expression patterns in one study, as drugs targeting this epigenetic modification, such as 5-azacytidine and decitabine, can reverse age-related neurodegeneration [124]. However, a later study reported the opposite effects, as exposure to 5-azacytidine for one day during development caused neurodegeneration in newborn mice and led to neurobehavioral impairments in adult animals [125]. The therapeutic potential of these and other epigenetic drugs for the treatment of brain pathologies has very recently been reviewed [126].

Histone methylation and acetylation guide the differentiation of NSCs into various neural cell types, including neurons and glial cells [127]. In contrast to the reversible and dynamic nature of acetylation, which is primarily linked to the expression of specific genes, histone methylation is characterized by its stability and potential involvement in the long-term maintenance of specific genomic areas [127]. Histone methylation is critical for the regulation of neurodevelopmental processes, synaptic plasticity, and the formation of long-term memories [128]. In particular, lysine methylation is a direct contributor to epigenetic inheritance and H3K4me has been found to promote transcriptional activation, while H3K9me is related to transcriptional suppression [127] (Table 1). Histone acetylation, particularly at genes associated with synaptic plasticity, plays a critical role in memory formation and the ability of neurons to strengthen or weaken their connections [129]. Previous research demonstrated that there was an elevation in histone acetylation inside the hippocampus following training, in contrast to untrained control subjects, and a reduction in histone acetylation was observed in other brain areas, such as the cortex (reviewed in [129]). Lysine deacetylase (KDAC) inhibitors, such as trichostatin A (TSA) and sodium butyrate (NaB), have been shown to augment long-term potentiation (LTP). Additionally, the administration of NaB through systemic injection has been demonstrated to enhance memory in vivo. The administration of TSA through intrahippocampal injection immediately following the learning process leads to improvements in long-term memory, while having no impact on short-term memory. This finding suggests that histone acetylation plays a crucial role in the consolidation of memory. Research has also indicated that the administration of NaB can promote the consolidation of long-term memory in response to mild stimuli and prolong the persistence of long-term memory. Therefore, it can be inferred that histone acetylation plays a significant part in the process of long-term memory formation [129]. Remarkably, histone acetylation, which has been associated with the establishment of long-term memory and synaptic plasticity, may take place at many lysine residues located within the four core histone proteins. It is worth noting that alterations in histone acetylation are associated with AD and may serve as potential diagnostic and therapeutic targets [130].

Histone phosphorylation is associated with synaptic plasticity and learning, contributing to the regulation of immediate early genes in response to neuronal activity, which is crucial for memory consolidation [131]. Histone H2B ubiquitination recruits H3K4me3 and plays a role in the regulation of several genes involved in neurodevelopment and synaptic plasticity [132]. Histone sumoylation contributes to the epigenetic regulation of genes involved in neuronal differentiation and synaptic plasticity by modulating N-methyl-D-aspartate (NMDA) receptors and L- and N-type voltage-gated calcium channels [133]. Crotonylation is another epigenetic modification that has been demonstrated in NSCs. This type of epigenetic mark is involved in NSC self-renewal and differentiation (by protecting pluripotency factors), as well as telomere protection [101].

ncRNAs, including miRNAs, regulate gene expression at synapses, influencing synaptic plasticity and learning processes [134]. Among the regulated genes is cAMP response element binding protein 2 (CREB2), which is crucial for long-term synaptic plasticity. The identification of distinctive DNA non-coding regulatory sequences that are important in brain cell differentiation, maturity, and plasticity has also been made possible by genome-wide analyses of epigenetic changes. Genomic enhancer elements are brief DNA regulatory sequences that bind transcription factors and work with gene promoters to increase transcriptional activity. This mechanism regulates gene expression programs crucial for determining the fate and function of neurons and is linked to many brain disease states [135]. Neurons are mostly rich in enhancers, which undergo bidirectional transcription to generate non-coding enhancer RNAs (eRNAs) and underlie dynamic gene expression patterns and cell-type specificity [135]. A list of references on the enhancers linked to neuronal development can be found in Supplementary Table S1 from [135].

### Dynamic Regulation of Epigenetic Modifications in Response to Environmental Factors

DNA methylation is a stable epigenetic modification; however, it can be dynamically altered in response to environmental factors such as diet, stress, toxins, and early-life experiences [136]. Studies have, e.g., shown that maternal diet during pregnancy can lead to changes in DNA methylation patterns in offspring, affecting long-term health outcomes [137].

Histone modifications can also be dynamically regulated in response to environmental factors, a phenomenon known as epigenetic plasticity. This process allows the genome to adapt to changing conditions and underscores the interaction between genes and the environment. Positive environmental factors, such as cognitive stimulation and physical activity, can promote histone modifications associated with synaptic plasticity and learning. Studies in rodents have shown that environmental enrichment can lead to increased histone acetylation and improved cognitive function [138]. On the other hand, exposure to drugs and environmental toxins also affects histone acetylation [139]. Chronic exposure to addictive substances, such as cocaine, can lead to changes in histone acetylation patterns in reward-related brain regions, contributing to addiction-related behaviors [140]. Histone methylation can also be dynamically regulated by stressors, including physical and psychological stress. Stress-induced changes in histone methylation can affect gene expression patterns in several areas of the brain [141], and chronic stress can lead to alterations in histone methylation in genes associated with mood regulation, contributing to the development of mood disorders [142].

## 3. Epigenetic Modifications in the Aging Brain

The decline in physiological functions that characterize aging is particularly apparent in the brain, which is mainly populated by postmitotic neurons that cannot be renewed and are, therefore, at risk of alterations leading to neurodegenerative disorders and/or neuronal death. One of the hallmark structural changes in the aging brain is a decrease in brain volume (atrophy), particularly in regions associated with memory and higher-order cognition, such as the hippocampus and the prefrontal cortex, which are critical for recall and executive function. While not as dramatic as in neurodegenerative diseases, the physiologically aging brain, particularly in selected regions, experiences some degree of downregulation of genes associated with synaptic and mitochondrial function, neuronal loss, and impaired microglial function, which can all contribute to the cognitive deficits observed in older people [143,144]. More specifically, a series of epigenetic modifications occur at the *Bdnf* gene, resulting in reduced levels of BDNF mRNA in the hippocampi of aged mice [145,146]. In addition, an altered NAD^+^/NADH ratio impacts the function of NAD^+^-dependent HDAC in aged neurons [147]. The activity of sirtuins declines gradually with age due to the drop in NAD^+^ levels in cells. SIRT1 can control axonal development, synaptic processes related to cognitive function, and synaptic plasticity in aged individuals [147,148,149]. Reduced SIRT1 function in aged neurons could hinder cognitive abilities in older individuals. Hippocampal CA1 neurons lacking Sirt1 exhibited reduced synaptophysin levels, poor LTP, and a decreased dendritic density [149]. Sirt1-deficient animals exhibit decreased CREB levels, leading to impaired CREB binding to BDNF and potentially causing reduced BDNF levels in the brain [150]. SIRT1 influences the creation of connections between neurons and their ability to change to control the process of memory formation. Another prominent feature of brain aging is the ability of dendritic spines to change their structure. Age-related reductions in spine number and maturity, as well as changes in synaptic transmission, may be a direct result of abnormal neural plasticity that affects the aged brain [151].

In addition, age-related alterations in the white matter, including demyelination and reduced integrity of white matter tracts, can lead to slowed information processing and cognitive decline [152].

Functional changes comprise cognitive decline with a reduction in processing speed, working memory, which is responsible for temporarily holding and manipulating information [153], episodic memory, which involves the ability to recall specific events and details [154], and changes in attention, including a reduced ability to filter out irrelevant information, which can affect task performance [155]. Other functional alterations disturb neurotransmitter systems, particularly the decline in dopamine and acetylcholine levels [156], and functional connectivity patterns within the brain’s networks, altering information processing and integration [157].

Aging has an obvious impact on neurological disorders, being the primary risk factor for neurodegenerative disorders such as AD and PD. Recent research has shown that epigenetic changes, specifically modifications to histones and DNA, play a pivotal role in the aging process and the development of age-related neurological conditions [158]. Aging also increases susceptibility to stroke and cerebrovascular diseases due to vascular changes, including reduced cerebral blood flow and the development of small vessel disease [159]. There are several ways in which epigenetic modifications can contribute to age-related cognitive decline. As mentioned in the previous section, the influence of epigenetic marks on the expression of genes associated with neuroplasticity and synaptic function may be effective in old age, leading, broadly speaking, to the impairment of brain structure and function [160]. For example, alterations in histone acetylation and DNA methylation are associated with AD and may serve as potential diagnostic and therapeutic targets [130]. Likewise, H3K27cr has been observed in AD to regulate exocytotic mechanisms of amyloid β clearance [161], and H4K12la is specifically activated in plaques of the 5XFAD mouse [30]. Epigenetic modifications in the aging brain can also serve as biomarkers for predicting age-related cognitive decline and the risk of developing neurodegenerative disorders [162]. Epigenetic clocks, which estimate biological age based on DNA methylation patterns, have shown promise in this regard [163]. Epigenetic clocks have, indeed, revealed that the epigenetic age of brain tissues can differ from chronological age and that at least certain areas of the brain may undergo accelerated aging [164]. Remarkably, single-cell transcriptomes from the neurogenic area of the subventricular zone in mice spanning various ages were recently obtained, and using regression models based on single cells, both the chronological and biological age (calculated from brain stem cell proliferation capability) were calculated, concluding that these aging clocks can be applied to several groups of mice, different brain areas, and other animal species [165]. In addition, heterochronic parabiosis (blood pairing of mice of dissimilar ages) and exercise were found to counteract transcriptome aging in neurogenic areas, although through distinct mechanisms to show that these clocks may be used to measure transcriptome rejuvenation. Therefore, epigenetic clocks have been linked to longevity and age-related health outcomes, and further research in the field may provide insights into the mechanisms underlying healthy aging [166].

An important feature of epigenetic modifications is that they are reversible, making them attractive targets for therapy [167]. Therefore, developing drugs or interventions that can modify epigenetic marks may offer avenues for slowing down the aging process or mitigating age-related neurodegenerative diseases [126]. Also of importance is that epigenetic changes in the aging brain are influenced by environmental factors, including diet, physical activity, and stress. Thus, understanding how these factors impact epigenetic modifications can inform lifestyle interventions that promote healthy brain aging [168].

Oxidative stress [169], chronic inflammation [170], changes in chromatin remodeling [171], dysregulation of the enzymes involved in histone regulation [172], senescent cells [173], and telomere shortening [174] are among the several factors that may contribute to histone epigenetic changes in the aging brain.

### 3.1. Contribution of Oxidative Stress to Epigenetic Changes in the Aging Brain

Oxidative stress is a prominent factor in the aging process and a hallmark of aging. Among its several consequences, oxidative stress may lead to different types of epigenetic changes, from DNA methylation to histone modifications and non-coding RNA profiles that can influence gene expression and contribute to age-related neurodegenerative conditions. Overall, oxidative-stress-induced DNA damage can impair the enzymes responsible for maintaining the epigenetic marks, leading to their dysregulation [175]. DNA damage can trigger changes in histone modifications, including increased histone H3K9 acetylation, which is associated with DNA repair processes [176]. Several effects have been described because of oxidative stress, including the aberrant methylation of CpG sites, resulting in DNA hypomethylation or hypermethylation. This can, in turn, affect the expression of genes involved in neuroprotection, synaptic plasticity, and inflammation [177]. Oxidative stress can also disrupt the balance of histone modifications. For instance, increased levels of oxidative stress may reduce acetylation and promote deacetylation, leading to the transcriptional repression of neuroprotective genes [178]. Oxidative stress can, likewise, influence the expression of non-coding RNAs, including miRNAs and lncRNAs. These non-coding RNAs can regulate the expression of genes associated with neurodegenerative processes [179].

### 3.2. Contribution of Inflammation to Epigenetic Changes in the Aging Brain

Inflammation is a central feature of aging-related neurodegenerative diseases, and it is increasingly being recognized as a contributor to epigenetic changes in the aging brain. Epigenetic modifications, including DNA methylation and histone acetylation, can regulate the expression of pro-inflammatory genes and trigger a vicious circle that contributes to the sustained activation of inflammatory pathways [180].

Alterations in DNA methylation patterns may affect the regulation of genes involved in immune responses, oxidative stress, and neuroinflammation [181]. Chronic inflammation sustained by pro-inflammatory cytokines, such as tumor necrosis factor α (TNFα), can lead to histone modifications that promote gene expression changes associated with inflammatory responses [182]. Thus, inflammation can increase histone acetylation at pro-inflammatory gene promoters, with sustained activation of inflammatory pathways [142]. Inflammatory processes can also alter the expression of miRNAs that target genes involved in neuroinflammation and neurodegeneration [183].

Another source of inflammation in the aging brain derives from senescent cells. Cells undergo a process known as cellular senescence, in which they alter their normal phenotype in response to stress and enter a prolonged cell cycle arrest state accompanied by a distinctive secretory phenotype [184], referred to as senescence-associated secretory phenotype (SASP), with the secretion, among others, of pro-inflammatory cytokines, growth factors, matrix-remodeling enzymes, and miRNAs. Additionally, senescent cells exhibit an altered morphology and proteostasis, a decreased propensity to undergo apoptosis, impaired autophagy, the accumulation of lipid droplets, and increased activity of senescence-associated-galactosidase (SA-gal). It is worth noting that SASP components can influence epigenetic changes such as DNA methylation, chromatin remodeling, and histone post-translational modifications in nearby cells [185], and that senolytic drugs selectively targeting and eliminating senescent cells have the potential to reduce inflammation and oxidative stress [151].

### 3.3. Contribution of Mitochondrial Dysfunction to Epigenetic Changes in the Aging Brain

Mitochondrial dysfunction, including mitochondrial stress, increased oxidative damage, and reduced ATP production, can lead to alterations in DNA methylation patterns. These changes may affect the regulation of genes involved in energy metabolism, oxidative stress responses, and neuronal survival [186]. Mitochondrial dysfunction can also lead to the altered expression of non-coding RNAs, including miRNAs and lncRNAs. Dysregulated non-coding RNAs can target genes involved in mitochondrial biogenesis, oxidative stress responses, and neuronal maintenance [187]. Epigenetic modifications regulate the expression of mitochondrial genes. These modifications can affect the efficiency of mitochondrial energy production and oxidative stress responses [188]. Mitochondrial dysfunction can lead to the release of mitochondrial-derived signals, such as reactive oxygen species (ROS) and mitochondrial DNA fragments. These signals can influence epigenetic changes in nearby cells, including neurons, leading to altered gene expression patterns [189].

### 3.4. Other Factors Contributing to Epigenetic Changes in the Aging Brain

Age-related changes in chromatin remodeling complexes can influence histone modifications. For instance, reduced activity of ATP-dependent chromatin remodeling complexes can lead to changes in histone acetylation and methylation patterns [190]. Likewise, the age-related dysregulation of enzymes responsible for adding (writers) or removing (erasers) histone modifications can result in imbalanced histone marks. For example, altered activity of HATs or histone deacetylases (HDACs) can affect histone acetylation levels [191]. In keeping with these observations, small molecules targeting epigenetic enzymes, such as DNMTs and HDACs, are being investigated as potential interventions to reverse age-related epigenetic changes and restore youthful gene expression patterns [192]. As mentioned previously, cellular senescence may induce epigenetic changes such as alterations in DNA methylation [193], histone-associated epigenetic mechanisms [194], chromatin remodeling [195], and non-coding RNA expression [195]. Senescence-associated epigenetic changes interact with the networks that regulate senescence and result in different phenotypes of cell senescence. Telomere shortening, a characteristic of aging, can also trigger chromatin alterations and changes in histone modifications at telomeric regions. These alterations can affect gene expression near telomeres [196]. This is because telomeres and subtelomeres, i.e., the regions of transition between chromosome-specific DNA and the telomere, possess histone and DNA modifications that are also highly concentrated in constitutive heterochromatin regions, such as pericentric heterochromatin [197]. Telomere shortening to a critically low length results in the development of epigenetic abnormalities at mammalian telomeres and subtelomeres [198]. These abnormalities are characterized by reduced levels of histone and DNA methylation, as well as elevated levels of histone acetylation. Age-related disorders such as accelerated aging syndromes are marked by extremely short telomeres, which might impact the epigenetic conditions of telomeres and subtelomeres [199].

## 4. Histone Modifications and Brain Aging

Recent research has shed light on the specific histone modifications linked to brain aging (Table 3). Studies have shown that a decrease in histone acetylation, particularly at genes associated with memory and synaptic plasticity, is associated with cognitive decline in aging individuals [200]. Age-related changes in histone methylation patterns have been observed in the brains of older individuals, and alterations in methylation at specific genes are linked to neurodegenerative diseases and cognitive decline [201]. We recently showed that, in the old mouse brain, histone H2AXγ phosphorylation is associated with caspase-dependent cell death and abortive cell cycle re-entry [202,203]. Developing small molecules that selectively target specific histone modifications, such as H3K4me3 or H3K27me3, may allow for precise modulation of gene expression relevant to cognitive function and neuroprotection [204]. It is worth mentioning that most studies examine bulk brain tissue, which may mask cell-type-specific epigenetic changes. Investigating histone modifications at the cellular level, especially in specific neuronal subtypes, can provide further insights into their roles in brain aging. In addition, much of the focus of current research has been on promoter regions, but understanding the role of histone modifications in enhancers, non-coding RNAs, and other non-coding regions is crucial for a comprehensive view of epigenetic regulation in brain aging [135].

As mentioned, the dysregulation of histone modifications has been implicated in neurodegenerative diseases such as AD and PD, which display a typical old-age onset. However, the exact role of histone modifications in disease pathogenesis and progression is not fully understood [215]. Studies have revealed alterations in histone modifications in the brains of individuals with AD. These changes include global reductions in histone acetylation levels and alterations in histone methylation patterns [200]. In AD, reduced histone acetylation, particularly at genes associated with memory and synaptic function, is linked to cognitive decline, and aberrant histone acetylation and methylation patterns are associated with disease progression [216]. Epigenetic drugs, such as histone deacetylase inhibitors (HDACIs), have shown promise in preclinical studies for their ability to reverse cognitive deficits and reduce amyloid-beta levels in animal models of AD [216]. The use of HDACIs to restore histone acetylation levels is, thus, a potential therapeutic approach to mitigating cognitive deficits in AD [217]. Aberrant histone methylation patterns are associated with tau pathology, one of the hallmarks of AD. Histone methylation marks have been found at specific tau gene promoters, affecting tau protein expression [218]. Targeting the HMTs involved in tau regulation could, thus, represent another potential therapeutic strategy to reduce tau pathology.

Emerging evidence suggests that epigenetic dysregulation, including histone modifications, also contributes to the pathogenesis of PD. These changes can affect gene expression patterns in the brain, influencing dopaminergic neuronal function and survival [219]. Altered histone acetylation patterns have been observed in animal models and the post-mortem brains of PD patients. These changes can affect the expression of genes involved in neuroinflammation and mitochondrial dysfunction, contributing to PD pathogenesis [220]. Sirtuin 1 (SIRT1), a histone deacetylase, plays a crucial role in regulating aging-related processes. SIRT1 can be activated through compounds like resveratrol, which may promote neuroprotection and cognitive function [221]. As in the case of AD, targeting HDACs to modulate histone acetylation levels is being explored as a potential therapeutic approach for PD [222]. Interestingly, aberrant histone methylation patterns have been linked to alpha-synuclein aggregation, a hallmark of PD, as they influence the expression of genes associated with alpha-synuclein metabolism and protein clearance [220]. Epigenetic modulators, such as HDACIs and HMT inhibitors, have shown promise in preclinical models of PD. These compounds can mitigate neuroinflammation, enhance protein clearance mechanisms, and protect dopaminergic neurons [223]. HDACIs such as vorinostat and valproic acid were reported to increase histone acetylation, promoting gene expression associated with synaptic plasticity and memory formation [224].

## 5. Epigenetic Clocks and Their Relevance for Aging

Epigenetic clocks have gained significant attention in the field of aging research due to their precision in predicting an individual’s biological age, which may differ from their chronological age [4]. DNA-methylation-based clocks, such as the Horvath and Hannum clocks, use specific patterns of DNA methylation at CpG sites to estimate biological age. These clocks have been validated in various tissues and populations [225]. Epigenetic clocks have been associated with health outcomes, including the risk of age-related diseases, such as cardiovascular disease and cancer, as well as overall mortality [226]. Epigenetic clocks can also measure “age acceleration,” which indicates whether individuals are aging faster or slower than expected based on their chronological age. Epigenetic clocks may serve as valuable tools for assessing the effectiveness of anti-aging interventions. They can be used to monitor changes in biological age in response to lifestyle modifications or medical treatments [227], as factors such as diet and exercise can influence age acceleration [166]. Despite their promise, there are challenges and debates surrounding the use of epigenetic clocks, including their biological interpretation and accuracy. Ongoing research aims to refine these clocks and enhance their predictive power [228].

## 6. Conclusions

The brain epigenetic landscape is emerging as a very important factor in the regulation of brain structure and function from development to old age. Although much progress has been made in understanding the roles of the main epigenetic modifications in the brain under normal and pathological conditions, most molecular changes have been discovered using biochemical and immunochemical approaches that have not always permitted linking these changes to specific brain areas and/or cell types. This will be the primary challenge in future research aiming to further proceed toward the translational use of these results in clinical practice.

## Figures and Tables

**Figure 1 ijms-25-03881-f001:**
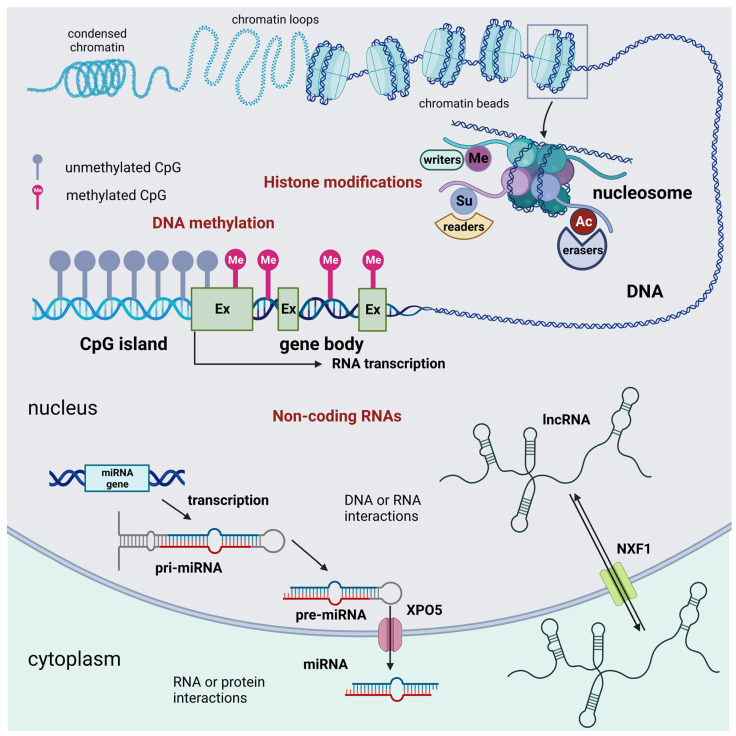
Schematic representation of the three main groups of epigenetic modifications in mammalian cells: DNA methylation, histone epigenetic modifications, and non-coding RNAs. DNA methylation acts as an off-switch to block translation and occurs at CpG sites that are observed across the genome. Methylation can occur in intergenic regions, CpG islands, and the gene body. Nevertheless, CpG islands that are considered normal exhibit a lack of methylation throughout all stages of development. This lack of methylation enables the transcription of the specific gene, provided that the necessary transcription factors are present, and the chromatin structure is accessible to these factors. Histone modifications are chemical alterations, which can have profound effects on gene expression and, consequently, various cellular processes. These modifications form an epigenetic code that imparts a distinct feature on chromatin architecture. The enzymes that catalyze these modifications can be classified as writers, readers, and erasers. Writers are enzymes that are responsible for the acetylation, methylation, phosphorylation, ubiquitination, sumoylation, lactylation, and crotonylation of histones. Among them, KMTs and HATs are of relevance. Readers are responsible for the recognition of the epigenetic marks on histones. Among readers are the readers of methyl- and acetyl-lysine residues. Among erasers are histone demethylases and deacetylases. Non-coding RNAs are divided into short and long non-coding RNAs. For simplicity, only the miRNA generation pathway is represented. Non-coding RNAs can interact with DNA, RNA, and protein molecules to modulate gene transcription, contribute to RNA inhibition or degradation, or serve as molecular guides, scaffolds, or decoys for specific proteins, such as transcription factors. These many functions occur either in the nucleus or the cell cytoplasm. Abbreviations: AC = acetylation; Ex = exon; lncRNA = long non-coding RNA; Me = methylation, demethylation, or trimethylation; miRNA = microRNA; NXF1 = nuclear RNA export factor 1; pre-miRNA; precursor miRNApri-mi RNA = hairpin-containing primary transcripts; Su = sumoylation; and XPO5 = exportin 5. Created with BioRender.com.

**Table 3 ijms-25-03881-t003:** Epigenetic marks in the old brain. Aminoacidic residues are indicated by one-letter notation [71]. For abbreviations see the list at the end of the paper.

Epigenetic Mark	Biological Effects	Brain Region	Target
**Reduction of H3K9ac**	Lowered expression of key genes to neuronal and synaptic development Decrease in age-related memory and learning capacity	Hippocampus	IEGs [205]
**Reduction of H3K14ac**		Hippocampus	IEGs [205]
**Reduction of H3K27ac**		Prefrontal cortex Hippocampus	GATA3, BDNF [146]
**Reduction of H4K12ac**		Hippocampus	Synaptic function-related genes [144]
**Increase in H3K9me2**	Aging	Cerebral cortex Hippocampus	Excitatory neurons [206]
**Increase of H3K9me3**	Reduction in dendritic growth and stability	Cerebral cortex Hippocampus	BDNF [207]
	Memory deficit	Hippocampus	BDNF [208] IEGs [209]
	Learning and memory ability decline	Brain tissue from AD patients and mouse models	Mitochondrial function-related genes [210]
**Increase of H3K4me2**	Increased expression of related stress response proteins and inducing cognitive impairment	Prefrontal cortex	Stress-related genes [211]
**Increase of H3K27me3**	Activation of stress and immune inflammation	Brain	Stress-related genes [212,213]
**Reduction of H3K36me3**	Impaired memory function	Cerebral cortex Hippocampus	BDNF [214]

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
