# Peer review of "An Overview of the Epigenetic Modifications in the Brain under Normal and Pathological Conditions"

_ijms, 2024, doi:10.3390/ijms25073881_

Round 1
Reviewer 1 Report (Previous Reviewer 2)
Comments and Suggestions for Authors
The resubmission of the manuscript entitled ‘An overview of the epigenetic modifications and their relevance in the normal and pathological brain’ by Lossi et al. is a revised and updated review. For this version, the original authors recruited a colleague who, according to the 'Author Contribution', did nothing worthwhile to mention. (Furthermore, who is X.X. who was responsible for visualization?) Obviously, this statement should be reviewed and corrected.
The revised and resubmitted manuscript has improved in many ways. The authors responded to the criticism with several corrections, clarifications, and additions (including about 120 new references, they more than doubled the list). The figures are informative and pleasing to the eyes; I noticed that they mention dimethylated or trimethylated marks in Figure 2. The Tables are useful, and now include extensive referencing, helping the reader enormously.
Most of my original major problems were addressed and many of the minor problems that I listed were corrected.
Minor problems that still exist:
1) Do not list the affiliation three times, you are all from the same place.
2) Line 81: For Me, monomethyl, dimethyl, and trimethylations should be mentioned.
3) lines 81-82: perhaps pri-miRNA and pre-miRNA should be explained.
4) Line 100: differential
5) Line 717: Who is X.X.?
6) Please, use a spell/grammar-checker program!
Comments on the Quality of English Language
Some editing of English language is necessary.
Author Response
We are grateful to the reviewer for her/his appreciation of our efforts. We have corrected all minor problems as indicated.
Reviewer 2 Report (Previous Reviewer 1)
Comments and Suggestions for Authors
The review is overall improved. There is still the "pathological brain" left in the title. Furthermore, I do not agree with the authors definition of an epigenetic modification but there is no general consensus in the field. In strict terms, meiotic and later mitotic modifications, which are not related to DNA sequence were considered as epigenetic. The inclusion of any kind of phenomenon leading to alterations in gene expression is not very productive and also not well defined. If you consider non-coding RNAs as epigenetic modifications, the same would apply for RNA-binding proteins, hypoxic transcript stabilisation, RNA hybrids etc. At least please mention the reference according to your point of view. If you would follow for instance the Adrian Bird definition, it is totally different from the NIH or the consensus from Cold Spring Harbour 2008
Comments on the Quality of English Language
English is fine, only some minor errors left.
Author Response
We are very pleased that the reviewer found substantial improvement in this paper version.
We apologize for not having corrected the title. We believed that the observation referred to the same expression in the text (that was amended). The title was changed in An Overview of the Epigenetic Modifications in the Brain under Normal and Pathological Conditions.
In light of the comment about the inclusion of non-coding RNAs among the types of epigenetic modifications, we have added the following at the beginning of the subparagraph
- Types of epigenetic modifications
Today there is no consensus in the field regarding the definition of epigenetic modifications. In his Nature paper of 2007 entitled Perceptions of Epigenetics Adrian Bird emphasized that for epigeneticists, there is no obvious 'epigene' [9]. He proposed to define epigenetics as the structural adaptation of chromosomal regions to register, signal, or perpetuate altered activity states by this means implicitly depicting epigenetic markings as reacting rather than proactive. Later, in 2008 a different definition was proposed in a meeting on chromatin-based epigenetics hosted by the Banbury Conference Center and Cold Spring Harbor Laboratory, defining an epigenetic trait as a stably heritable phenotype resulting from changes in a chromosome without alterations in the DNA sequence [10]. Thus there is no consensus as to whether or not non-coding RNAs could be regarded as participating in epigenetics. Yet Shelley Berger and colleagues, in providing their view and interpretation of the proceedings at the meeting, have considered non-coding RNAs among epigenetic initiators [10]. Since non-coding RNAs, as discussed later in this paper, have been described as participating in several brain epigenetic modifications we will consider them as a third category of these modifications.
We thank the reviewer for her/his observation.
Reviewer 3 Report (New Reviewer)
Comments and Suggestions for Authors
The article is well written and presents a good review of the topic. It also presents 2 tables, which help to better understand the biological effect of histones, and Proteins acting on DNA methylation in the nervous tissue. However, I think it could be substantially improved by explaining the way this research was carried out, what keywords were used to find these articles and not others on the topic? In this sense, I think a chapter should be included saying how the research strategy was carried out?
In this sense, I consider that the article should be approved after this chapter has been inserted.
Author Response
Thank you very much for the good feedback on the paper.
As requested we have added a short description of our search strategy in a new subparagraph of the introduction.
- Literature search strategy
If one searches for the string “epigenetic changes in the brain” in PubMed filtering for reviews or systematic reviews and the last ten years more than 1,200 papers are retrieved. In writing this paper we aimed to focus on the studies describing histologically and functionally the main epigenetic modifications of the mammalian brain during development, adulthood, and old age. We have also taken into consideration the more important findings related to the most diffused neurodegenerations, i.e. Alzheimer’s disease (AD) and Parkinson’s disease (PD). We used PubMed as a starting database but also made direct Internet searches of exact phrases (in double quotes) related to the most important topics to be addressed. The keywords used were “epigenetic”, “brain”, “neurons”, “development”, “aging”, and “mammals”. These terms were first identified in the PubMed database and their synonyms were recognized in the thesaurus. Variations in search terms were also considered. The search was primarily focused on the last 10 years, but older relevant papers were also included.
This manuscript is a resubmission of an earlier submission. The following is a list of the peer review reports and author responses from that submission.
Round 1
Reviewer 1 Report
Comments and Suggestions for Authors
Lossi & Merighi review “Epigenetic modifications and their relevance in the normal and pathological brain”. The focus on DNA methylation, histone modifications, and non-coding RNAs. The current version of the manuscript clearly lacks detail. Most of the citations are reviews on the different aspects of their topic and rarely original observations are cited or described. The descriptions are in general mostly common place and not detailed in terms of the known molecular events. Some examples are given below:
Line 52: I have a problem with the concept: non-coding RNAs are not epigenetic modifications.
In the methylation introduction, the authors focus mainly on CpG methylation. What about adenine?
Table 1: “ZBTB4 and ZBTB38 exhibited high expression levels in the brain” is not a function.
Line 115ff: Please add additional recent references instead of 1 ten years old review please.
The same applies for Refs 17 and 18 etc. At least the key papers should be cited (PMID: 16724059, 18539123).
Line 245: Again, please read and cite the original papers, e.g. 5-azacytidine might have the opposite effects of what is described here PMID: 27594097.
Line 329ff: “physiologically aging brain, particularly in selected regions, experiences some degree of neuronal loss, which can contribute to the cognitive deficits observed in the elderly [43].” Please try to be more precise; statements like this are not informative. Same for 363ff.
Line 346ff: The sentence is incomplete.
Line 379: How senescent cells and telomere shortening are supposed to contribute to histone epigenetic changes?
Line 530: there is no “pathological brain” please re-phrase.
Please use the same character size for main text and conclusion.
Line 592: There is no experimental work in the review.
Comments on the Quality of English Language
English is fine.
Reviewer 2 Report
Comments and Suggestions for Authors
The manuscript entitled “Epigenetic modifications and their relevance in the normal and pathological brain” by Lossi and Merighi is a reasonable attempt to review some current research on brain epigenetics. Albeit the Figures are informative and pleasing to the eyes, and the Tables are useful, there are some major as well as several minor problems with this manuscript.
Major problems
1) There are many reviews about epigenetics in relation to neuronal functions in health and disease. A quick search in PubMed for ((epigenetic[Title/Abstract]) AND (brain[Title/Abstract])) AND (review[Publication Type]) brings up about 2757 review publications. This tells a lot about the extreme need for another one. I wonder how the authors would define their goal with this manuscript. My advice would be to narrow your search field for a review that is more focused to a particular aspect of brain epigenetics.
2) I would recommend adding “A brief introduction” to the title to harmonize with the current scope of the manuscript: “A brief introduction to the epigenetic modifications and their relevance in the normal and pathological brain”.
3) The authors should add further references to the places indicated below. It is especially important for the Tables. For example, the columns entitled “Function” in Table 1, or “Biological effects” in Tables 2 and 3, need extensive referencing.
4) Some paragraphs are based on a single reference; this approach suggests that the authors rely too heavily on other reviews and not primary sources. See, for example, lines 115-133 or 190-202. Further, some paragraphs read as textbook samples (see, for example, the subchapter on DNA methylation (lines 72-84 with two references!) or Histone modifications (line 97-99 with a single reference!)). This is not how serious reviews are made.
5) The authors need to work on the manuscript: the text has a few of annoying problems from an unfinished sentence to confusing statements about funding… A thorough grammar check and a solid stylistic refurbishing (preferably by a native speaker) are needed.
Minor problems
6) Line 11: Perhaps there is no need for the word “basic”.
7) Line 17: Please define the meaning “developmental” in this context. If the primary meaning is for ontogenetic development, then the authors are advised to use references from this area as well. The current list of literature is overwhelmingly about adult brain tissue. If it is used for the development/progression of a disease, it should be emphasized.
8) Lines 48-50: This sentence needs references.
9) Lines 55-56: Correctly: “…DNA methylation, histone epigenetic…”
10) Lines 72-84: Here, an entire paragraph is with only one reference that points to Table 1 which has no reference at all. Please provide some.
11) Line 85: Please note in the legend of the table that the abbreviations are at the end of the text.
12) Line 87: Di- or trimethylations should also be mentioned and referenced. I would also add them to Figure 1 where currently only single methylations are seen.
13) Lines 104-146: There is only one reference (ref. 7) for practically two pages. These subchapters should be considered important, so they should be thoroughly referenced.
14) Line 141: Please provide references for the biological effects, and explanations for the single letter abbreviations for amino acids.
15) Line 142: Correctly: Ac…, La
16) Lines 200-202: An entire paragraph for the description of ncRNAs. Unfortunately, their function in epigenetic processes is missing. Please provide details with references.
17) Line 213: Ref. 20 is in superscript.
18) Line 347: The sentence is incomplete.
19) Lines 378-381: This paragraph needs a few references.
20) Line 462: Wrong use of a square bracket.
21) Lines 472-473: In the Table: what are “ac”, “acc”, and “a” for the different histone Lys modifications?
22) Line 589: Who is X.X.?
23) Lines 592-593: There is no original experimental work reported in this manuscript. What does this sentence mean? Does it mean data published in refs. 84 and 85 were funded by local grants? There is no need to list them here.
24) Line 597: The reference list includes a note: “From NLM” or “From NLm”; are these important? Further, only two references (1, 67) have links instead of the commonly used DOIs. References should be listed with uniform formatting.
Comments on the Quality of English Language
The authors need to work on the manuscript: the text has a few of annoying problems from an unfinished sentence to confusing statements about funding… A thorough grammar check and a solid stylistic refurbishing (preferably by a native speaker) are needed.
Reviewer 3 Report
Comments and Suggestions for Authors
This paper is an adequate review of the state-of-the-art knowledge on epigenetics and its effects altering brain function, structure and development. For the clinician's interest the implications of dysregulation of the histone in the genesis of certain neurodegenerative disorders such as Alzheimer's and Parkinson's Disease opens possibilities of eventual development disease modifying therapies. The brief discussion on historical antecedents is welcome as well as the list of abbreviations.
The general concept is that from the mechanistic point of view, the diverse epigenetic mechanisms result from environmental or external factors inducing the complex (molecular changes described in the paper), with consequential dysfunctional effects. I feel comments on specific factors inducing epigenetic changes, i.e. smoking and DNA methylation, are missing to illustrate your otherwise clear review. Your comments in this respect will be appreciated.